# The Fate of *Yersinia pseudotuberculosis* in Raw Fermented Meat Products

**Radka Hulánková *** and **Irena Svobodová**

Department of Animal Origin Food & Gastronomic Sciences, Faculty of Veterinary Hygiene and Ecology, University of Veterinary Sciences Brno, Palackeho tr. 1946/1, 61242 Brno, Czech Republic; svobodovai@vfu.cz
* Correspondence: hulankovar@vfu.cz

**Abstract:** *Yersinia pseudotuberculosis* is a foodborne pathogen with an animal reservoir, thus being able to spread via contaminated meat. The survival of *Y. pseudotuberculosis* during the ripening and storage of artificially contaminated spreadable fermented sausage (Teewurst) and dry fermented sausage was studied, with initial counts of 8 log, 6 log, and 3 log CFU/g. While the pathogen was completely inhibited in all batches of dry fermented sausage after 4 d of ripening and was thus absent in the final product, it survived much better in the spreadable sausage, characterized by a higher pH and fat content. The counts in the Teewurst final product (after 2 d of ripening) dropped from 8 log, 6 log, and 3 log CFU/g to approx. 6.3, 2.4, and 1.4 log CFU/g, respectively. For the initial concentrations 6 log and 3 log CFU/g, at least 1 out of six samples was still positive after 20 d of cold storage. On the other hand, in the batches with the highest initial counts (8 log CFU/g), all the samples were positive at the end of the experiment (37 d). The rapid decline in pH caused by the starter culture was an effective barrier for *Y. pseudotuberculosis* in dry fermented sausages, but the pathogen was able to persist in Teewurst.

**Keywords:** Teewurst; fermentation; ripening; water activity; food safety

## 1. Introduction

The genus *Yersinia* includes Gram-negative, facultative anaerobic bacteria that are found worldwide in a variety of environments [1]. The entire genus currently includes 28 species [2], of which only three are pathogenic. The best-known species is *Yersinia (Y.) pestis* (causative agent of plague), which is very similar to *Y. pseudotuberculosis* at the genetic level [3]. *Y. pseudotuberculosis* can also contaminate the food production environment, and its presence has been confirmed in water [4]. In situations of inadequate hygiene, improper sanitation or high primary contamination of raw materials, *Yersinia* spp. can contaminate meat and meat products and cause foodborne illness.

*Y. enterocolitica* and *Y. pseudotuberculosis* are significant foodborne pathogens that cause the gastrointestinal illness known as yersiniosis [2,5]. In humans, yersiniosis can appear in acute or subacute forms, and less commonly as a chronic infection. The acute form of yersiniosis is a typical gastroenteritis with abdominal pain, diarrhea, and fever, sometimes with vomiting, affecting mainly young children. The subacute form of yersiniosis presents similar gastrointestinal symptoms and fatigue. Possible complications include mesenteric lymphadenitis, which mimics appendicitis, or extra-intestinal manifestations like erythema nodosum and reactive arthritis. Chronic yersiniosis manifests as prolonged enterocolitis or as a post-infectious complication with large joint inflammation, erythema nodosum, thyroiditis, or rarely kidney inflammation. The majority of human yersiniosis

cases are sporadic; however, in patients with other comorbidities, immunodeficient patients, the elderly, or young children, yersiniosis may be more severe, and sequelae may occur. *Y. pseudotuberculosis* infections tend to be more severe and invasive than those caused by *Y. enterocolitica*, with the mesenteric lymph nodes, liver, and spleen being more frequently affected. Infections caused by *Y. pseudotuberculosis* are underestimated worldwide [5–8].

Both *Y. enterocolitica* and *Y. pseudotuberculosis* also cause diseases in animals with similar symptoms to yersiniosis in humans, i.e., fever, diarrhea, enteritis, and lymphadenitis. Various animals are affected, including livestock, pets, birds, rodents, and wild and zoo animals. Most infections are asymptomatic; therefore, animals can serve as a reservoir of *Yersinia* spp. *Yersinia* spp. may be transmitted from pigs and other livestock infected on the farm during slaughter to carcasses and organs [3,6,8]. The ability of *Yersinia* to grow at low temperatures is important for food safety. Therefore, in raw meat products, increased attention must be focused on other barriers to growth, such as the pH drop during fermentation or the reduction in water activity ($a_w$) during drying.

Raw fermented meat products represent a relatively wide range of products that are very popular in European countries. The principle of processing of raw fermented sausages is that they are aged in air-conditioned chambers without heat treatment. The whole process of aging involves microbial fermentation, which is how this type of salami obtained its name. Raw fermented meat products can be subdivided according to different criteria and countries of origin into further, smaller groups. According to the shelf life and water activity, they can be divided into dry sausages and undried sausages. Dry salami is popular not only in Europe (Hungarian, Italian, German and Greek salamis, Spanish chorizos, Czech Polican, Herkules, etc.) but also in the US (US-style pepperoni). The production process usually consists of several days of fermentation followed by several weeks of ripening and smoking. Their low pH (4.6–5.5) and low $a_w$ (<0.9) make them suitable for storage at room temperature (~25 °C). Undried sausages (e.g., Teewurst, coarse onion Mettwurst) are products that have not undergone significant ripening or drying. They have a limited shelf life, even when stored under proper refrigeration (<7 °C). The pH value is below 5.3, and the $a_w$ is usually high (0.95–0.97) [9]. In the production of raw fermented meat products, attention is paid to the low microbial contamination of the raw material, strict recipe control, and curing conditions that ensure product safety. Nevertheless, this group of meat products poses a risk, particularly due to *Yersinia*'s ability to grow at refrigeration temperatures. Performing a risk assessment for foodborne diseases requires information on the behavior of pathogens under different processing and handling conditions that occur throughout the food chain. Data on the growth and survival of *Y. pseudotuberculosis* in raw fermented meat products have been nonexistent so far. The main aim of this study was to assess the survival of *Y. pseudotuberculosis* in two different types of raw fermented meat products during processing and storage (Teewurst and a dry fermented sausage).

## 2. Materials and Methods

### 2.1. Preparation of Inoculum

A mixture of five wild strains of *Yersinia pseudotuberculosis* was prepared. The strains were isolated from the caecal content or tonsils of wild boars during experiments conducted at our department in previous years [10]. All the strains belonged to serotype O:1, biotype 1.

The strains were kept frozen at −70 °C. Before the experiment, they were subcultured twice on TSA agar (Oxoid, Basingstoke, UK) at 30 °C for 24 h. A suspension of each strain in sterile distilled water was prepared using a photometer (Densilameter II, Erba Lachema, Brno, Czechia), adjusted to the 10.0 McFarland turbidity standard (~$3 \times 10^9$ cells/mL). The suspensions of each strain were mixed in equal ratios and, if necessary, diluted to obtain

200 mL of inoculum at a concentration of approximately 8 log, 6 log, or 3 log CFU/g of raw batter.

## 2.2. Preparation of Spreadable Fermented Sausage (Teewurst)

Two batches of approx. 4 kg of raw batter were prepared for each concentration of *Y. pseudotuberculosis*. The spreadable fermented sausage (Table 1) was prepared from pork shoulder, pork backfat, nitrite curing salt (Maso-Profit, Prague, Czechia), seasoning mix SM 11 (Raps, Kulmbach, Germany), and starter culture Biostart Duo 50 (Raps, Germany). The raw material was pre-cut using a 6 mm grinder plate and finely comminuted with other ingredients in a bowl cutter. After the preparation of the raw batter, 200 mL of inoculum was added and mixed well. The final batter was stuffed into polyamide casing of 40 mm in diameter (Devro, Slavkov, Czechia). In total, 26 sausages per batch were manufactured, each weighing about 140 g. The products were placed into a smoking chamber (TITAN S029, Maurer, Munich, Germany) and treated with cold smoke for 2 days at relative humidity (RH) 80–90% and 18–20 °C, and then transferred to a chilling room and kept at 3 ± 1 °C.

**Table 1.** List of ingredients for the fermented sausages.

| Ingredient | Teewurst | Dry Fermented Sausage |
|---|---|---|
| Pork shoulder | 1.6 kg | 1.6 kg |
| Beef neck | - | 1.2 kg |
| Pork backfat | 2.4 kg | 1.4 kg |
| Nitrite curing salt | 88 g | 88 g |
| Seasoning mix | 32 g | 32 g |
| Starter culture | 2 g | 2 g |
| Total | 4.122 kg | 4.322 |

## 2.3. Preparation of Dry Fermented Sausage

Two batches of approx. 4 kg of raw batter were prepared for each concentration of *Y. pseudotuberculosis*. The dry fermented sausage (Table 1) was manufactured from pre-frozen meat and fat (beef neck, pork shoulder, and pork backfat), nitrite curing salt (Maso-Profit, Czechia), seasoning mix Rohwurst Polican (Raps, Germany), and starter culture Biostart Duo 50 (Raps, Germany).

The raw material was pre-cut before freezing using a 6 mm grinder plate. All the ingredients were then ground and mixed in a bowl cutter. After preparation, 200 mL of inoculum was added and mixed well in the bowl cutter. The batter was filled into natural pork casings 32/34 mm (Maso-Profit, Czechia). From each batch, 24 products weighing approx. 150 g were made. The sausages were treated with cold smoke in a smoking chamber (TITAN S029, Maurer, Germany). The conditions were 24 °C, RH 92–94% for 30 h; followed by 22 °C, RH 88–90% for 20 h; 20 °C, RH 84–86% for 34 h; and 17 °C, RH 82% for 84 h. The final product was transferred to a chilling room and kept at 3 ± 1 °C.

## 2.4. Physicochemical, Microbiological, and Statistical Analyses

The water activity and pH were measured daily. The ripening period of the dry fermented sausage was finished when the water activity reached ≤0.930, in accordance with Czech legislation [11].

The water activity ($a_w$) value was measured using an AquaLab 4te Duo machine (METER Group, Pullman, WA, USA) according to ISO 21807, with a resolution of 0.0001. The pH values were measured using a pH meter WTW 3110 with a resolution of 0.001 (Xylem Inc., Rye Brook, NY, USA), equipped with a puncture glass electrode Hamilton Double Pore (Hamilton Comp., Reno, NV, USA) and calibrated using pH standards 4.0 and

7.0 at 25 °C. A temperature sensor NTC30 (Mettler Toledo, Greifensee, Switzerland) was used during the measurement of samples for automatic temperature correction.

Before inoculation, a sample from each batch was taken and analyzed for presence of *Yersinia* spp. (with a negative result).

For a microbiological analysis, the sampling of the spreadable fermented sausage took place on Day 0 (raw batter after inoculation), Day 2 (final product), Day 10, Day 20, Day 30, and Day 37. The sampling of the dry fermented sausage took place on Day 0 (raw batter after inoculation), Day 2, Day 4, Day 7 (final product), Day 9, and Day 11. In the preliminary experiment, the sampling period for dry fermented sausage was up to 48 d, but since there was an absence of *Yersinia* in the final product, we decided to shorten the intervals and monitor the changes during ripening more closely. Three samples per batch were taken on each sampling day. In total, 216 samples were analyzed.

For the detection of *Y. pseudotuberculosis*, 25 g of sample in a sterile blender bag with lateral filter (VWR International, Radnor, PA, USA) was homogenized in a Stomacher (SMASHER®, bioMérieux, Marcy-l'Étoile, France) with 225 mL of Peptone Sorbitol Bile (PSB) broth (M941, Himedia, India) and incubated at 30 °C for 24 h. The enrichment was followed by streaking on Cefsulodin-Irgasan-Novobiocin (CIN) agar (M843, Himedia, Mumbai, India) and cultivation at 30 °C for 24 h.

The enumeration of *Y. pseudotuberculosis* was performed by homogenizing 10 g of sample with 90 mL of PSB broth. After resuscitation for 2 h at laboratory temperature, serial dilutions were made when necessary, and 0.1 mL was spread in duplicate on CIN agar and incubated at 30 °C for 24 h. The number of typical colonies formed was counted and reported as log CFU/g for each sample.

The results of the analyses are reported as mean values ± standard error of the mean (SE). A mixed model ANOVA with Tukey HSD test was applied to analyze the fixed effects and their interaction (initial concentration and days after manufacture), with the batch (A or B) as the random effect. The two types of meat products were compared in a separate model, as the comparison was possible only for Day 0 and Day 2. A level of significance of 0.05 was used. Statistical analyses were performed using the statistical program STATISTICA v. 7.1 (StatSoft, Tulsa, OK, USA). For statistical purposes, values below the limit of detection (LOD) were replaced with a value of 1.0 log CFU/g.

## 3. Results

Two different types of raw fermented meat products were manufactured and analyzed to assess the survival of *Y. pseudotuberculosis.* Not surprisingly, both the initial concentration and the length of storage (including their interaction) were highly statistically significant factors ($p < 0.001$). In Teewurst, *Y. pseudotuberculosis* survived throughout the storage period (for 37 d) when the initial concentration was around 8 log CFU/g (Table 2). The numbers gradually decreased throughout the storage by 4.7 log CFU/g. During the fermentation period, the counts dropped by 1.3 log CFU/g. Another highly significant decrease was noted between Day 30 and Day 37, suggesting that the numbers would drop below the LOD not long after.

However, in sausages with lower initial concentrations, the numbers of *Y. pseudotuberculosis* declined much more quickly, falling below the LOD on Day 10 for most of the batches. The counts in the final product were approx. 1.4 and 2.4 log CFU/g for initial concentrations of 3 log and 6 log, respectively. Even when the counts dropped below 2 log CFU/g, *Y. pseudotuberculosis* was still viable in at least one out of six samples and could be recovered by plating after pre-enrichment.

**Table 2.** Survival of *Y. pseudotuberculosis* (mean ± standard error) in raw spreadable fermented sausage (Teewurst).

| Inoculum | Production Process | | | Storage | | |
|---|---|---|---|---|---|---|
| | Day 0 | Day 2 | Day 10 | Day 20 | Day 30 | Day 37 |
| 3 log CFU/g | 2.89 ± 0.10 [a] 6/6 * | 1.43 ± 0.20 [b] 6/6 | ≤1.00 ± 0.00 [b] 3/6 | ≤1.00 ± 0.00 [b] 1/6 | ≤1.00 ± 0.00 [b] 0/6 | ≤1.00 ± 0.00 [b] 0/6 |
| 6 log CFU/g | 5.51 ± 0.11 [c] 6/6 | 2.43 ± 0.08 [d] 6/6 | 1.11 ± 0.11 [b] 6/6 | ≤1.00 ± 0.00 [b] 2/6 | ≤1.00 ± 0.00 [b] 0/6 | ≤1.00 ± 0.00 [b] 0/6 |
| 8 log CFU/g | 7.61 ± 0.11 [e] 6/6 | 6.30 ± 0.08 [f] 6/6 | 5.52 ± 0.07 [c] 6/6 | 5.03 ± 0.10 [g] 6/6 | 4.77 ± 0.03 [g] 6/6 | 2.90 ± 0.17 [a] 6/6 |

* Number of positive samples using horizontal method of detection. [a–g] Different letters in superscript indicate statistically significant differences ($p < 0.05$).

In dry fermented sausage, the survival of *Y. pseudotuberculosis* was clearly more hampered by the proliferation of the starter culture. In batches with initial concentration of 3 and 6 log CFU/g, no sample was positive on Day 2 of manufacture (Table 3). In batches starting with 8 log CFU/g, complete inactivation was delayed by only 2 d. The counts on Day 2 differed significantly between Teewurst and dry fermented sausage in batches starting with 8 and 6 log CFU/g ($p < 0.001$).

**Table 3.** Survival of *Y. pseudotuberculosis* (mean ± standard error) in dry fermented sausage.

| Inoculum | Production Process | | | | Storage | |
|---|---|---|---|---|---|---|
| | Day 0 | Day 2 | Day 4 | Day 7 | Day 9 | Day 11 |
| 3 log CFU/g | 2.84 ± 0.06 [a] 6/6 * | ≤1.00 ± 0.00 [b] 0/6 | ≤1.00 ± 0.00 [b] 0/6 | ≤1.00 ± 0.00 [b] 0/6 | ≤1.00 ± 0.00 [b] 0/6 | ≤1.00 ± 0.00 [b] 0/6 |
| 6 log CFU/g | 5.78 ± 0.07 [c] 6/6 | ≤1.00 ± 0.00 [b] 0/6 | ≤1.00 ± 0.00 [b] 0/6 | ≤1.00 ± 0.00 [b] 1/6 | ≤1.00 ± 0.00 [b] 0/6 | ≤1.00 ± 0.00 [b] 0/6 |
| 8 log CFU/g | 7.96 ± 0.07 [d] 6/6 | 2.41 ± 0.15 [e] 6/6 | ≤1.00 ± 0.00 [b] 0/6 | ≤1.00 ± 0.00 [b] 0/6 | ≤1.00 ± 0.00 [b] 0/6 | ≤1.00 ± 0.00 [b] 0/6 |

* Number of positive samples using horizontal method of detection. [a–e] Different letters in superscript indicate statistically significant differences ($p < 0.05$).

The differences between the two types of fermented products could be attributed to variations in pH, and water activity ($a_w$), and fat content. The first two d of fermentation resulted in the most pronounced decrease in pH for both products ($p < 0.001$). The pH values dropped lower in dry fermented sausage than in Teewurst (Figure 1), which can be explained by differences in temperature and RH during manufacture, chemical composition, and diameter of the products. The pH values in dry fermented sausage gradually decreased to around 4.5, which is the minimum value enabling growth under otherwise optimal growth conditions [12].

The decrease in water activity due to drying was also much more pronounced in dry-fermented sausage compared to Teewurst (Figure 2), which would also act as a barrier for *Yersinia* proliferation. The highest decrease was observed between Day 4 and Day 7 ($p < 0.001$), corresponding with the onset of more intensive drying. The value 0.93, stipulated by Czech legislation as the maximum for dry fermented sausages [11], was achieved on Day 7. However, since *Yersiniae* were already completely absent on Day 4 with $a_w$ being 0.95–0.96, it is clear that water activity was not the key parameter in the elimination of *Y. pseudotuberculosis* in our experiment.

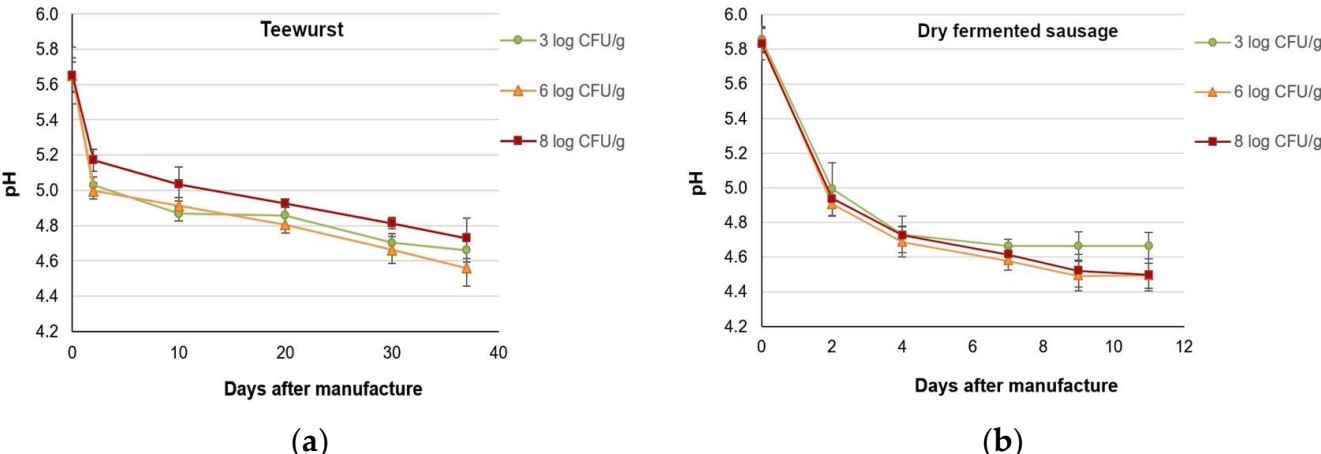

**Figure 1.** Values of pH (mean ± standard error) during manufacture and storage of raw spreadable fermented sausage Teewurst (**a**) and dry fermented sausage (**b**).

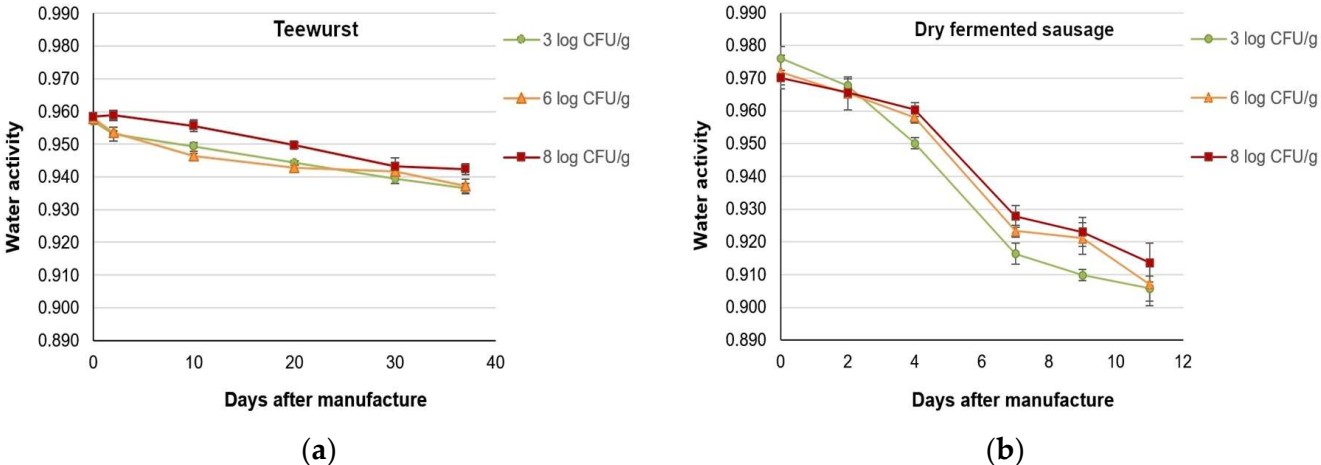

**Figure 2.** Values of water activity (mean ± standard error) during manufacture and storage of raw spreadable fermented sausage Teewurst (**a**) and dry fermented sausage (**b**).

## 4. Discussion

*Y. pseudotuberculosis* is a poorly studied pathogen, and no data on its survival in (fermented) meat products have been published so far. Even data on its viability and growth in raw meat are limited [13,14]. However, multiple authors have studied the fate of *Y. enterocolitica* in fermented sausages. Asplund et al. [15] noted the positive effect of nitrites and starter culture on the inhibition of *Y. enterocolitica*. Starter culture also significantly decreased the relative abundance of *Yersinia* spp. in Chinese Sichuan sausages [16]. In another study, inoculated *Y. enterocolitica* (approx. 5 log CFU/g) was completely eliminated in Turkish dry fermented sausage (Sucuk) by *Lactobacillus sakei* and *Pediococcus acidilactici* by the third day of fermentation [17]. Lindqvist and Lindblad [18] noted that *Y. enterocolitica* survived poorly in Swedish fermented sausage, with a much higher inactivation rate in comparison with *Listeria monocytogenes* or *Escherichia coli*. On the other hand, in the study by Ivanovic et al. [19], *Y. enterocolitica* was eliminated in a dry fermented sausage only after 25 d, with initial counts of 6 log CFU/g. However, despite using a starter culture, the pH values were 4.8 or higher, and water activity was above 0.93 throughout the entire ripening and storage period. Interestingly, *Y. enterocolitica* was also found in a fermented Sardinian sausage in two from nine samples (22%) with a final pH value of 5.29 [20].

*Y. pseudotuberculosis* survived much better in raw spreadable sausage compared to dry fermented sausage, which was particularly evident at the highest level of inoculation, with counts dropping during the first 2 d of fermentation by approx. 1.3 and 5.5 log CFU/g in spreadable and dry sausage, respectively. This finding is consistent with results from other studies. Sperandii et al. [21] reported the good survival of *Y. enterocolitica* in traditional salsiccia fresca (coarse raw Italian pork sausage) stored at 8 °C for 7 d, with increasing numbers of the pathogen during prolonged storage (20 d). The German Federal Institute for Risk Assessment (BfR) conducted a study on *Yersinia enterocolitica* and found that, unlike in salami produced with a starter culture, *Y. enterocolitica* was detectable throughout the shelf life of spreadable sausages Teewurst and Mettwurst [22]. In the study of Lorencova et al. [23], inoculated *Mycobacteria* could be detected in Teewurst during 4 w of storage, but no viable cells were found in dry fermented sausage (final product). Böhnlein et al. [24] reported that Shiga toxin-producing *E. coli* (STEC) was inhibited during the production of dry fermented salami but not in Teewurst. In general, pathogenic bacteria can be reduced by 2–4 log in semi-dry and dry sausages, but only by 1 log or less in spreadable sausages [25]. This enhanced survival could be related to the higher pH values and higher fat content in Teewurst. It was reported that a low pH and high fermentation temperature improved the inhibition of STEC, while the larger diameter of casing and higher fat content prolonged its survival [26]. Teewurst is a finely cut (meat-in-fat emulsion) meat product, usually containing 35–55% of fat, with a minimum being 25% to keep the product spreadable [27]. The fat may protect the bacteria and enhance their survival. It is also important to note that the term "Teewurst" may refer to various products, including "cooked Teewurst" in the USA or products without a starter culture characterized by pH~6.0 [28], whereas the German products typically have a final pH of 5.3–5.5 (semi-acidified) or even 4.9–5.0 [24,27].

The German type of Teewurst has been previously implicated in STEC infections [29]. In a German study, the consumption of Mettwurst, a coarsely cut raw sausage similar to Teewurst, was found to be a significant risk factor in contributing yersiniosis [30]. Since there is very limited information on *Y. pseudotuberculosis*, it is hard to estimate the realistic level of contamination in meat products. If we extrapolate from *Y. enterocolitica* counts in meat, Fredriksson-Ahomaa et al. [31] reported numbers lower than 1 log CFU/g in fresh pork, but the counts increased up to 4.3 log CFU/g during cold storage. Messelhäusser et al. [32] found the level of contamination of pork to be between −1.4 log and 5.4 log CFU/g. Thus, the lowest artificial contamination level 3 log CFU/g used in our study is not completely impossible, although it would depend on the meat freshness, etc. Since from the initial 3 log contamination *Y. pseudotuberculosis* survived in 1 out of 6 samples (16.7%) for 20 d and in 50% of samples for 10 d, its presence in Teewurst during its shelf life cannot be completely excluded even in the case of lower initial counts. Further studies could shine more light on the real contamination levels and effect of storing temperature on survival and growth, but generally, Teewurst should be considered as a risky food not only for *Y. enterocolitica*, but also for *Y. pseudotuberculosis*, and appropriate control measures and detection methods should be engaged to also include this pathogenic *Yersinia* species.

Although the infectious dose is supposed to be high, such as 8 log CFU [33], the contamination level of 3–4 log CFU in several outbreaks in Finland has been implied [33,34]. Thus, even relatively low numbers may pose a risk for human health, especially for risk groups, such as pregnant women, the elderly, small children or individuals affected by immunosuppression or polymorbidities.

## 5. Conclusions

The results of this study indicate that *Y. pseudotuberculosis* cannot survive the processing of dry fermented sausage manufactured with a starter culture and with a final pH value below 4.7. Although the bacteria survived better in raw spreadable sausage (with higher pH, aw, and fat content), the risk for healthy adults is very low since the pathogen was able to survive for 20 d, but only in very low numbers (<2 log CFU/g), and was not able to multiply. Higher numbers were detected throughout the storage period in the most heavily inoculated batches, but it should be noted that the initial level of inoculation of 8 log CFU/g does not correspond with the naturally low levels of contamination of fresh meat and carcasses by *Yersinia* spp. [31,32,35–37]. However, for certain population groups, such as the elderly or small children, short-fermented spreadable products made from raw pork could still pose a health risk, which is higher compared to other ready-to-eat meat products such as dry fermented salami.

**Author Contributions:** Conceptualization, R.H.; methodology, R.H.; formal analysis, R.H.; writing—original draft preparation, R.H. and I.S.; writing—review and editing, R.H. and I.S.; visualization, R.H.; funding acquisition, R.H. and I.S. All authors have read and agreed to the published version of the manuscript.

**Funding:** This research was funded by the Institutional Support for the University of Veterinary Sciences Brno.

**Institutional Review Board Statement:** Not applicable.

**Informed Consent Statement:** Not applicable.

**Data Availability Statement:** Data available in a publicly accessible repository Mendeley Data, https://data.mendeley.com/datasets/t73mbbnrzk/2 (accessed on 20 March 2025).

**Conflicts of Interest:** The authors declare no conflicts of interest.

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
