# Peer review of "The Fate of Yersinia pseudotuberculosis in Raw Fermented Meat Products"

_applsci, doi:10.3390/app15105324_

Round 1

Reviewer 1 Report

Comments and Suggestions for Authors

Paper: “The fate of Yersinia pseudotuberculosis in raw fermented meat products”

by Hulánková and Svobodová., Applied Sciences

The current study aimed to assess the survival of Y. pseudotuberculosis in two different types of raw fermented meat products during processing and storage (Teewurst and a dry fermented sausage). I find the topic interesting for Applied Sciences, however, in its current form, the manuscript has some unclear points in the text that should be changed. In general, the reading of the paper is clear in various parts of the text, but has some gaps. For these reasons, I recommend revising the manuscript, my decision is minor revision.

For these reasons, kindly suggest that the manuscript be revised.

In particular, the authors of the proposed manuscript should focus on the following comments:

-Comments:

  • The abstract is unclear I would ask you to rewrite it for better understanding.
  • In line 25, after environments, add the reference.
  • In line 83, I ask you to cite the work where the isolation of these strains used in this study is shown.
  • Line 88 express the concentration of McFarland turbidity standards for greater clarity.
  • In line 94/109, it would be more appropriate to list the sausage preparation quantities of both productions in the table.
  • In line 109 you have listed the ingredients for 4 kg of mixture. However, I don't add up in relation to the kg of the various ingredients used. Could there be an error in the accounts? the sum does not add up to 4 kg.
  • Include a reference to Czech legislation stating the following in line 124.
  • In line 134 I would put analysis and not examination.
  • In line 142 -143 Reports the brand name of both the sterile plastic bags and the stomacher.
  • In line 143 It would be more appropriate to write the acronym PSB. I think it is phosphate saline buffer.
  • In line 144 It would be more appropriate to write the acronym CIN agar.
  • In line 147 and in line 149 You could omit the manufacturers of substrates and solutions, because you mentioned them earlier.

  • Overall, some changes are needed to improve the content of the article. In my opinion, the paper cannot be published in Applied Sciences; my response is a minor revision.

Author Response

Comments 1: The abstract is unclear I would ask you to rewrite it for better understanding.

Response 1: The abstract was slightly modified to make it more readable.

Comments 2: In line 25, after environments, add the reference.

Response 2: Reference added – L26: [1] Fàbrega, A.; Vila, J. Yersinia enterocolitica: pathogenesis, virulence and antimicrobial resistance. Enferm Infecc Microbiol Clin. 2012, 30, 24–32.

Comments 3: In line 83, I ask you to cite the work where the isolation of these strains used in this study is shown.

Response 3: Reference added – L86: [10] Hulankova, R. Higher Resistance of Yersinia enterocolitica in Comparison to Yersinia pseudotuberculosis to Antibiotics and Cinnamon, Oregano and Thyme Essential Oils. Pathogens 2022, 11, 1456.

Comments 4: Line 88 express the concentration of McFarland turbidity standards for greater clarity.

Response 4: The concentration and its determination is now detailed on L90-91.

Comments 5: In line 94/109, it would be more appropriate to list the sausage preparation quantities of both productions in the table.

Response 5: Table 1 was added with the list of ingredients.

Comments 6: In line 109 you have listed the ingredients for 4 kg of mixture. However, I don't add up in relation to the kg of the various ingredients used. Could there be an error in the accounts? the sum does not add up to 4 kg.

Response 6: Thank you for pointing this out. The amount and ratio of ingredients are correct, but the final weight of the batter was not exactly 4.0 kg. The text was revised to „approx. 4 kg“ (L97, L110) and the final weight of the batter is stated in Table 1.

Comments 7: Include a reference to Czech legislation stating the following in line 124.

Response 7: Reference added – L128: [11] Decree No. 69/2016 on requirements for meat, meat products, fishery and aquaculture products and products thereof, eggs and products thereof. Czech Collection of Laws. FAOLEX No. LEX-FAOC174728.

Comments 8: In line 134 I would put analysis and not examination.

Response 8: Corrected (L138).

Comments 9: In line 142 -143 Reports the brand name of both the sterile plastic bags and the stomacher.

Response 9: Corrected (L146-148).

Comments 10: In line 143 It would be more appropriate to write the acronym PSB. I think it is phosphate saline buffer.

Response 10: Corrected (L148).

Comments 11: In line 144 It would be more appropriate to write the acronym CIN agar.

Response 11: Corrected (L150).

Comments 12: In line 147 and in line 149 You could omit the manufacturers of substrates and solutions, because you mentioned them earlier.

Response 12: Corrected (L153-154).

Reviewer 2 Report

Comments and Suggestions for Authors

1. Given that the initial contamination levels (up to 8 log CFU/g) do not reflect natural contamination scenarios, how might the results differ under more realistic, low-level inoculation conditions?
2. The types of starter cultures used and their specific impact on pH reduction and microbial inhibition in dry fermented sausages should be more detailed.
3. Were the cold storage conditions (e.g., temperature and humidity) for Teewurst reflective of typical consumer behavior, and how might variations in home refrigeration affect pathogen survival?
4. What was the rationale for using six samples per condition, and is this sample size sufficient to capture batch variability and support broader conclusions?
5. Does the noncovalent interaction impact the Yersinia pseudotuberculosis in raw fermented meat products? ACS Appl. Polym. Mater. 2025, 7, 1459−1470 should be refered.
6. How do the survival characteristics of Y. pseudotuberculosis compare with those of other foodborne pathogens in similar meat matrices?

Author Response

Comments 1: Given that the initial contamination levels (up to 8 log CFU/g) do not reflect natural contamination scenarios, how might the results differ under more realistic, low-level inoculation conditions?

Response 1: The first problem is that given the very limited information on Y. pseudotuberculosis, it is hard to estimate the realistic level of contamination. If we extrapolate from Y. enterocolitica counts in meat, Fredriksson-Ahomaa et al. (2012) reported numbers less than 1 log CFU/g in fresh pork, but the counts increased up to 4.3 log CFU/g during cold storage. Messelhäusser et al. (2011) found the level of contamination of pork between -1.4 log  and 5.4 log CFU/g. Thus the lowest artificial contamination level 3 log CFU/g used in our study is not completely impossible, although it would depend on the meat freshness etc. Since from the initial 3 log contamination, Y. pseudotuberculosis survived in 1 out of 6 samples (16.7 %) for 20 d and in 50 % of samples for 10 d, its presence in Teewurst during the shelf-life cannot be completely excluded even in the case of lower initial counts. Further studies could bring more light on the real contamination levels and effect of storing temperature on survival and growth (see Comment 3 Response), but generally Teewurst should be considered as risky food not only for Y. enterocolitica, but also for Y. pseudotuberculosis, and appropriate control measures and detection methods should be engaged to include also this pathogenic Yersinia species.

Although the infectious dose is supposed to be high, such as 8 log CFU (Pärn et al., 2015), contamination level of 3 - 4 log CFU in several outbreaks in Finland has been implied (Jalava et al., 2006; Pärn et al., 2015). Thus even relatively low numbers may pose a risk for human health, especially for risk groups, such as small children (the most common age group affected by yersiniosis), pregnant women, the elderly or individuals affected by immunosuppression or polymorbidities.

Comments 2: The types of starter cultures used and their specific impact on pH reduction and microbial inhibition in dry fermented sausages should be more detailed.

Response 2: We are well aware that the composition of the starter culture would be beneficial for the quality of our manuscript and could enable further discussion. Unfortunately, the manufacturer keeps this as his trade secret and was not willing to share this information.

Comments 3: Were the cold storage conditions (e.g., temperature and humidity) for Teewurst reflective of typical consumer behaviour, and how might variations in home refrigeration affect pathogen survival?

Response 3: Humidity plays no important role in Teewurst storage, as the polyamide casing itself significantly reduces evaporation. Storage temperature would be definitely more interesting to study, as the high temperatures in home refrigerators are notorious, with the reported mean air temperature 6.1 – 6.4 °C but maximum over 10 °C (Bonanno et al., 2024). The temperature selected in our study (3 °C) reflects the usual recommended storage temperature range from +1 to +5 °C (unpublished market survey). It would be interesting to find out if Y. pseudotuberculosis at a higher storage temperature can overcome the hurdles represented by low pH and low water activity and better survive or even multiply, or if the inactivation rate by lactic acid at higher temperature would be even higher, as proved for Y. enterocolitica in vitro and in fermented sausages by Lindqvist and Lindblad (2009).

Bonanno, L.; Bergis, H.; Gnanou-Besse, N.; Asséré, A.; Danan, C. Which domestic refrigerator temperatures in Europe? - Focus on shelf-life studies regarding Listeria monocytogenes (Lm) in ready-to-eat (RTE) foods. Food Microbiol. 2024, 123, 104595.

Comments 4: What was the rationale for using six samples per condition, and is this sample size sufficient to capture batch variability and support broader conclusions?

Response 4: The number of samples is a compromise, representing a sufficient amount of samples for statistical analysis while keeping the number of samples manageable regarding the costs and laboriousness of the analyses.

Comments 5: Does the noncovalent interaction impact the Yersinia pseudotuberculosis in raw fermented meat products? ACS Appl. Polym. Mater. 2025, 7, 1459−1470 should be refered.

Response 5: Unfortunately, we were not able to obtain the full version of the recommended article even through contacting the authors, so we were not able to assess its possible relation to our research. However, it seems that the publication “A Reproducible and Self-Repairable Ionic Skin with Robust Performance Retention Enabled by Modulating the Noncovalent Interactions“ aims more for non-covalent interactions in artificial hydrogel based on poly(acrylic acid-acrylamide. Non-covalent bonds are important in manufacture of fermented sausages in the process of gel formation from proteins, as the gel stabilizes the fat particles and is important for the texture of the product. Bacteria, especially lactic acid bacteria, have a significant role in the gel formation, so the bacteria and their metabolites impact the formation of non-covalent interactions, not vice versa (Hao et al., 2024).

Hao, S.; Qian, M.; Wang, Y.; Zhang, K.; Tian, J.; et al. Research progress on the gel properties of fermented sausage. Food Mater. Res. 2024, 4,  e007.       

Comments 6: How do the survival characteristics of Y. pseudotuberculosis compare with those of other foodborne pathogens in similar meat matrices?

Response 6: Since there is no other article on Y. pseudotuberculosis survival in fermented meat products, the whole discussion part is oriented at other pathogens, mostly Y. enterocolitica, but also Shiga toxin-producing E. coli. Our results generally do not differ from similar studies on the other foodborne pathogens, but it is not valid to compare exact numbers, since all the studies more or less differ in inoculation dose and changes in pH and water activity during ripening (e.g. L230-232). 

Reviewer 3 Report

Comments and Suggestions for Authors

This manuscript provides the systematic evaluation of the survival dynamics of Yersinia pseudotuberculosis in two types of fermented meat products. the results hold significant relevance for food microbiology and public health safety. The manuscript is well-structured, with detailed methodology and a discussion section that effectively integrates existing literature. However, certain aspects require refinement to enhance scientific rigor and readability.

  1. The potential pathogenicity of Y. pseudotuberculosis at low bacterial loads (e.g., presence of resuscitation or risk of residual toxin) needs to be further explored.
  2. Please mark the groups directly in the figure(Teewurst,dry fermented sausage).
Comments on the Quality of English Language

Ensure consistent reference formatting

Author Response

Comments 1: The potential pathogenicity of Y. pseudotuberculosis at low bacterial loads (e.g., presence of resuscitation or risk of residual toxin) needs to be further explored.

Response 1: As mentioned in the manuscript, Y. pseudotuberculosis is a poorly studied pathogen. Although the infectious dose is supposed to be high, such as 8 log CFU (Pärn et al., 2015), contamination level of 3 - 4 log CFU in several outbreaks in Finland has been implied (Jalava et al., 2006; Pärn et al., 2015). Thus even relatively low numbers may pose a risk for human health, especially for risk groups, such as small children (the most common age group affected by yersiniosis), pregnant women, the elderly or individuals affected by immunosuppression or polymorbidities.

The text has been added to the manuscript (L260-280).

Comments 2: Please mark the groups directly in the figure (Teewurst, dry fermented sausage).

Response 2: Corrected.

Comments 3: Ensure consistent reference formatting.

Response 3: Corrections were made where necessary throughout the Reference part.

Round 2

Reviewer 2 Report

Comments and Suggestions for Authors

The paper can be accepted as the quality has been improved.